# Evaluation of Contrast Flow Patterns with Cervical Interlaminar Epidural Injection: Comparison of Midline and Paramedian Approaches

**DOI:** 10.3390/medicina57010008

**Published:** 2020-12-24

**Authors:** Byeongcheol Lee, Sang Eun Lee, Yong Han Kim, Jae Hong Park, Ki Hwa Lee, Eunsu Kang, Sehun Kim, Jaehwan Kim, Daeseok Oh

**Affiliations:** Department of Anesthesia & Pain Medicine, Inje University Haeundae Paik Hospital, Busan 612-896, Korea; h00543@paik.ac.kr (B.L.); painlee@paik.ac.kr (S.E.L.); h00111@paik.ac.kr (Y.H.K.); h00150@paik.ac.kr (J.H.P.); tedy333@paik.ac.kr (K.H.L.); h00347@paik.ac.kr (E.K.); shottherbz@paik.ac.kr (S.K.); h80530@paik.ac.kr (J.K.)

**Keywords:** cervical vertebrae, injections, contrast media

## Abstract

*Background and objectives:* The purpose of this study was to compare and to analyze contrast spread patterns between the paramedian and midline approaches to cervical interlaminar epidural injection (CIEI). *Materials and Methods:* We retrospectively enrolled 84 CIEI cases that had been performed for unilateral cervical spinal pain from April 2019 to April 2020. After 3 mL of contrast had been injected into the epidural space, fluoroscopic images were obtained. The CIEI was divided into a midline (Group M, *n* = 42) and a paramedian (Group P, *n* = 42) approach by anteroposterior imaging. The P Group was classified into a more medial (Group Pm, *n* = 26) and a more lateral (Group Pl, *n* = 16) group. Using ImageJ on an anteroposterior image, we assessed the grayscale brightness ratio of the ipsilateral or contralateral side of the vertebral body as well as the intervertebral disc space one level just above the needle location. We identified the dispersion of contrast into the ventral epidural space. *Results*: The grayscale brightness ratio was significantly higher in Group P than in Group M (*p* < 0.001). The incidence of ventral epidural spread in Group M was 57.1% versus 88.1% in Group P, which was significantly different (*p* = 0.001). *Conclusions:* The fluoroscopic CIEI finding in the paramedian approach predominantly showed an excellent delivery of the injectate to the ipsilateral side in comparison to the contralateral side. This showed a greater advantage in delivery toward ventral epidural space as compared to the midline approach.

## 1. Introduction

Epidural injection of local anesthetics and corticosteroids increases the concentration of the treating agent within the epidural space to inhibit inflammation, resulting in decreased edema and reduced nociceptive afferent signaling [1]. Theoretically, the number of therapeutic injectate doses targeted to the painful pathologic lesion is important. The incidence of ventral epidural spread in the lumbar spine is strongly correlated with clinical improvement, based on one study performed [2], although the fluoroscopic findings of contrast spread and the clinical effect on the cervical spine have yet to be compared. Anatomically, transforaminal (TF) injections can better target the site of pathology in the spine. However, TF injection is not recommended at the cervical level, due to the risk of severe complications such as death, brain/spinal cord infarction, anterior spinal artery syndrome, or direct nerve injuries [3,4]. Therefore, interlaminar epidural injection at the cervical level is widely used to treat acute and chronic cervical spinal pain. The midline and the paramedian approaches are used for cervical interlaminar epidural injection (CIEI). The paramedian or a more lateral approach to the symptomatic side allows the spread of a more targeted and specific therapeutic injectate ipsilaterally and enables the spread of injectate throughout the ipsilateral epidural space [5]. The paramedian approach can replace a TF injection as the initial procedure for unilateral cervical radiculopathy by analyzing the clinical effectiveness of fluoroscopic CIEI [5]. We believe that the paramedian approach for CIEI is comparable to the TF approach in delivering the injectate into the ipsilateral and ventral epidural sides effectively. However, it has yet to be elucidated fully in clinical investigations. Previous subjective analyses of contrast medium spread in the cervical spine were highly variable [6,7,8,9,10]. Further comparisons of clinical effectiveness between the midline and paramedian approaches also yielded conflicting results [8,11]. To our knowledge, no consensus is available regarding the superiority of either approach for management of cervical spinal pain.

We tried to quantitatively compare the contrast spread pattern in the cervical epidural space between the midline and paramedian approaches by using the ImageJ software. The purpose of this study was to detail evidence showing the differences between the two approaches for contrast media and therapeutic injectate in reaching the suspected site of the origin of cervical pain during CIEI.

## 2. Materials and Methods

### 2.1. Study Population and Data

This study was approved by the Institutional Ethics Committee of Inje University Haeundae Paik Hospital, Republic of Korea (HP IRB 2020-05-003) and registered at https://cris.nih.go.kr (KCT0005587). We retrospectively reviewed the electronic medical records, radiologic studies, and anteroposterior (AP) and lateral or contralateral oblique (CLO) fluoroscopic images acquired from a single pain clinic center during intervention of 110 cases where CIEI had been performed from April 2019 to April 2020 because of cervical spinal pain. The patients received CIEI under fluoroscopic guidance, either via a midline (M) or a paramedian (P) approach. We confirmed on an AP view that the spinous process was equidistant from both pedicles. We identified the cranial spread of contrast media on a lateral view at more than one level above the needle tip location. We designated each approach as the relation between the final needle tip location and the vertical lines of the spinous process as seen on an AP view [11]. The midline approach was defined as being within the lateral margin of the spinous process from one side to the other. The paramedian approach was defined as being when the needle tip was positioned outside of the vertical lines of the spinous process on the ipsilateral side. We classified the paramedian approach as either more medial (Pm) or more lateral (Pl). Based on previous analysis, the area from the lateral margin of the spinous to the lateral margin of the interlaminar opening measured at its maximum width can be divided into two equal zones: Group Pm medially and Group Pl laterally [6,12]. We excluded a total of 26 cases for the following reasons: patients who had two CIEIs (10), previous cervical spine operation (6), asymmetric radiologic images (6), images where contrast media had spread to other areas around epidural space (3), and the insertion state of a spinal cord stimulator lead (1) (Figure 1).

### 2.2. Injection Technique

Two experienced pain physicians performed all procedures under fluoroscopic guidance. The strategy depends on the practitioner’s preference and technical challenges caused by the patient’s condition. We placed the patients in the prone position with the spine positioning system under the chest. AP and lateral views were obtained with a fluoroscopy unit (OEC 9900 Elite, GE OEC Medical Systems, Salt Lake City, UT, USA) to ensure that the interlaminar epidural space had been accurately determined. If the needle tip was obscured by the shoulder in the lateral view, a 45~50° CLO view was used. Following aseptic preparation and infiltration with 2% lidocaine, a 26-gauge needle was inserted into the skin surface. A 20-gauge 8 cm Tuohy needle with cephalad bevel orientation was advanced into the epidural space. A loss of resistance using saline was used to identify the epidural space under fluoroscopic guidance. If a loss of resistance was felt, 1.0 mL of contrast medium (Omnipaque, 300 mg/mL, GE Healthcare, Little Chalfont, UK) was injected to detect the epidural space after confirming negative aspiration. Subsequently, a total of 3 mL of contrast medium was injected to observe the spread pattern on AP and lateral or CLO views. The final fluoroscopic images of contrast medium spread were saved.

### 2.3. Image and Data Analysis

Two pain physicians reviewed all images on a picture archiving and communication system. The etiology was analyzed based on the patients’ medical records and a review of imaging studies, including CT and MRI. We analyzed the stored AP images that had been downloaded from Picture Archiving and Communication System (PACS) workstation with ImageJ software (Version 1.53, https://imagej.nih.gov/ij/, National Institutes of Health). An investigator performed a brightness scale analysis using ImageJ software. To study the image, we drew a rectangular region of interest (ROI) on the vertebral body and intervertebral disc space at the level of the ipsilateral space just above the needle, using the toolbar’s rectangular selection tool. We then drew another rectangular ROI of the same size that was generated automatically on the contralateral side, based on the spinous process. The ROI selection did not overlap with the vertical lines of the spinous process or the pedicle lines. The selected areas’ grayscale brightness value was calculated by using the analyze tool in ImageJ program (Figure 2). A grayscale brightness value of 0 indicates true black, and a value of 255 indicates true white. We identified the mean grayscale brightness value of each area. The mean value was determined by averaging the values from the ROI boxes on the image. If the contrast medium spreads in the ROI, the value shows a relatively lower level. However, it is difficult to determine by the grayscale brightness alone whether contrast media is present or not because it can be affected by various factors such as fluoroscopic settings and patient impact. Therefore, we only compared the ratio of the grayscale brightness value that had been determined by dividing the brightness value of the contralateral side with that of the ipsilateral side. Determination and selection of the ROI areas may be subjective because of some degree of individual bias. To reduce this bias in designating the ROI, two independent pain physicians participated in the computer-assisted analysis. The presence or absence of contrast spread into the ventral epidural space was also evaluated on lateral or CLO images (Figure 3).

### 2.4. Statistical Analysis

The data that are presented include the frequency and percentage for categorical variables and mean ± standard deviation (SD) for numeric variables. The differences in study participants’ characteristics were compared across subgroups by using the chi-squared test or Fisher’s exact test for categorical variables and an independent test or Mann–Whitney’s U test for continuous variables as appropriate. To check the normality of the data distribution, we used Shapiro–Wilk’s test. For data visualization, a boxplot with dots and a bar chart with error bars were also displayed. All statistical analyses were carried out by using SPSS 24.0, and *p* values of less than 0.05 was considered statistically significant.

## 3. Results

We included a total of 84 CIEIs that were performed in 84 patients, and the basic characteristics are listed in Table 1. This study included Group M (*n* = 42) and Group P (*n* = 42). Group P was divided into Group Pl (*n* = 16) and Group Pm (*n* = 26). Each group was comparable in age, sex, impression, and the cervical level of injection. The grayscale brightness ratio and the presence of ventral epidural spread between Group M and Group P were demonstrated (Table 2). The grayscale brightness ratio was significantly more pronounced in Group P (1.91 (1.50–2.35)) as compared to Group M (1.17 (1.10–1.41); *p* < 0.001). The presence of a ventral epidural spread was 24 (57.1%) in Group M and 37 (88.1%) in Group P. There were significant differences between Group M and Group P (*p* = 0.001). No significant difference in variables was observed between Group Pm and Group Pl (Table 3). As shown in Table 4 and Figure 4, a relatively high brightness ratio was observed in cases of the presence of ventral epidural spread (*p* < 0.001).

## 4. Discussion

We found that the contrast spread for the paramedian approach tended to be significantly greater on the ipsilateral side at the vertebral body and intervertebral disc space of just one level above the needle location than it was in the midline approach. The rate of spread to ventral epidural space in the paramedian approach was 88.1% versus 57.1% of the space, which was significantly different (*p* < 0.05). There can be detailed evidence, considering that the CIEI needle is clinically positioned to either the specific target level of the pathologic site or to the radicular pain source.

Anatomical examination of the lumbar epidural space revealed that there was a septum-like connective tissue called plica mediana dorsalis. This potential barrier in the midline of the posterior epidural space may restrict contrast flow or lead to unilateral spread [13,14]. Previous clinical studies described the various contrast patterns, based on intuitive interpretations, such as unilateral and bilateral spread in AP view during the performance of CIEI [6,7,8,10]. The contrast flow was found to spread evenly and bilaterally up the cervical spine in all cases where CIEI was performed in the midline [7]. The unilateral contrast spread was observed in 51% of the cases during paramedian approach [10]. A recent study reported a 100% predominantly bilateral spread on the midline approach and 100% predominantly ipsilateral spread in the paramedian approach [8]. We assumed it is visually difficult to determine the entire area of contrast dispersion via fluoroscopic imaging because of the relatively small differences in the proportion of contrast area bilaterally, suggesting possible ambiguity and subjective elements. We calculated the grayscale brightness using ImageJ software to evaluate the 3 mL contrast spread pattern on the ipsilateral and contralateral side in the AP view, based on computer-assisted quantitative analysis. Computer-assisted quantitative analysis of the contrast spread between the ipsilateral and the contralateral side with caudal epidurography has been reported [15]. The authors compared the spread of contrast between the two sides by counting the number of pixels within all the areas of contrast present, although the determination of all areas of contrast spread was also subjective to some degree. Grayscale brightness ratio analysis has been used as a tool for assessing the echo intensity to evaluate image quality between the nerve structure and its surrounding muscles, and it has demonstrated that grayscale analysis is a valid and reproducible method for measuring echo intensity [16,17]. We used it as a way to assess the differences in the proportion of contrast media within the cervical epidural space on fluoroscopic AP imaging. This method has the advantage of quantitative evaluation compare to previous studies. A difference of grayscale brightness shows an unequal distribution of contrast media on a specific region. This means that as the ratio increases, the asymmetrical dispersion increases. In our study, the paramedian approach showed a predominantly unilateral tendency to spread.

The fluoroscopic findings of the paramedian approach showed a higher grayscale ratio value and ventral epidural spread than did the midline approach. We found that the group showing ventral epidural spread demonstrated a relatively high value in the grayscale brightness ratio when a 3 mL volume of contrast medium had been given, regardless of approach methods. The more contrast spreads to the ipsilateral side, the more likely it is to reach to the ventral epidural space. Previous reports have described varying results regarding the incidence of ventral spread during CIEI [6,7,8,9,10], but these were not directly comparable. The various factors responsible could be the differences between approach techniques, contrast volumes, injection levels, unclear radiologic criteria, and underlying cervical pathologies in the study subjects. The contrast spread range was especially dependent on the contrast volume in the cervical epidural space, regardless of approach differences [9]. A reduced posterior epidural space in the cervical spine as compared to the lumbar spine may also contribute to a greater extent of blockage [18]. In our cases, the incidence of ventral epidural spread was 57.1% in Group M and 88.1% in Group P. We also predicted that a volume greater than 3 mL might ensure a higher incidence of ventral epidural spread into both groups. However, a larger volume can cause unwanted consequences by significantly increasing the rate of bilateral and longitudinal spread, or by excessive pressure in the cervical epidural space. There is no consensus on what the optimal volume of solution should be in consideration of clinical pathologic lesions. An increased volume of dispersion into the epidural space, such as steroid delivery to the targeted areas of pathology, may reduce the medication’s concentration. In other words, if the injectate spreads more to the ipsilaterally affected side, a reduced volume of injectate may be required in the pathologic site so that a greater concentration of medication can reach the targeted area.

We compared Group Pl with Group Pm, but the results did not show value in intentionally advancing the needle more laterally. Lee et al. [5] suggested that a slight paramidline needle placement is enough to deliver to the unilateral cervical epidural space. However, the approach to the most lateral epidural space has been shown to influence ventral epidural spread in the lumbar spine [19]. Choi et al. [20] recommend using a modified paramedian approach and advancing the needle more laterally in order to deliver the drug more efficiently to the lateral and ventral epidural space, as this is as effective as the TF approach. It seems that more research will be needed to determine whether the cervical epidural needle tip’s distance from the spinous process during the paramedian approach affects drug delivery to the ipsilateral pathological lesion.

We analyzed five cases that revealed limitations to ventral epidural spread, despite the paramedian approach. Although all cases showed variable brightness ratios, each value was lower than the average value of those cases that showed ventral epidural spread. We identified predominantly longitudinal or bilateral spread on fluoroscopic imaging, but it did not reach to the ventral epidural space (Figure 5). This is likely the result of anatomical restrictions or a barrier causing an increased resistance to the flow of contrast into the epidural space of the pathology side [15], and we suspect that it may be related to the pathological condition.

Our study had several limitations. First, this study was a retrospective design without a uniform approach strategy. This may have affected the results and caused some bias. Second, we did not analyze clinical outcomes according to which approach techniques were used. There have been clinical studies that compared the differences in treatment effectiveness between a midline and a paramedian approach to the cervical spine [8,11], yet previous studies have shown contradictory results regarding the clinical efficacy of the paramedian approach as compared to the midline approach. These reports did not analyze the relevant results of the spread pattern between the two approaches [8,11]. In view of this, further controlled studies that compare detailed fluoroscopic findings of contrast between the midline and paramedian approaches should be conducted, with consideration of clinical efficacy in the cervical spine. Third, there are likely to be limitations because the analysis of contrast spread was performed on two-dimensional images [15]. We did not consider the range of longitudinal spread between the two approaches, and thus we did not select for all areas of contrast medium spread because of technical limitations. The grayscale brightness ratio also may be affected by spread thickness variability. For example, a markedly high brightness ratio corresponds to an increased ipsilateral epidural thickness, such as epidural pooling or distension, with a relatively small amount of longitudinal spread. We predicted the spread pattern indirectly by analyzing contrast medium on a single level of vertebral body and intervertebral disc space, just above the needle location, although we did not evaluate the entire range of contrast spread.

## 5. Conclusions

In conclusion, when the CIEI needle is inserted toward the pathological side, the spread of contrast medium tends to be predominantly ipsilateral to the needle compared with the midline approach. Thus, the contrast spread to the lateral compartment of the ipsilateral cervical epidural space is promoted, and the paramedian approach eventually facilitates the entry of contrast medium into the ventral epidural space. This study provides evidence supporting the advantage of paramedian CIEI compared with the midline approach, resulting in the delivery of a more concentrated medication close to the ipsilateral pathologic site and ventral epidural space to treat unilateral cervical spinal pain. We suggest that the paramedian approach is more effective than the midline approach for the conservative management of unilateral cervical spinal pain.

## Figures and Tables

**Figure 1 medicina-57-00008-f001:**
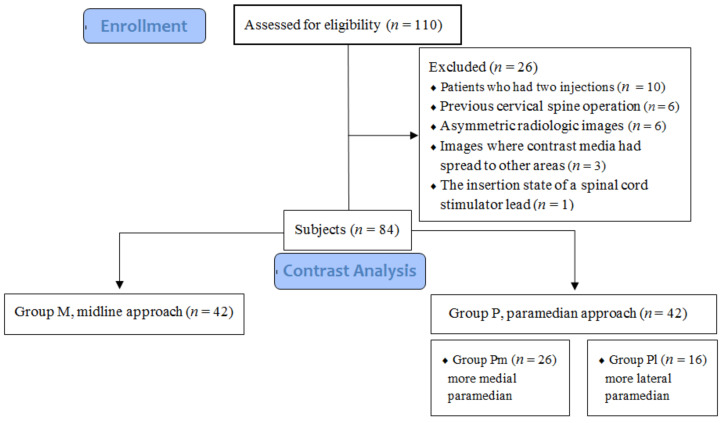
Flow diagram of the study.

**Figure 2 medicina-57-00008-f002:**
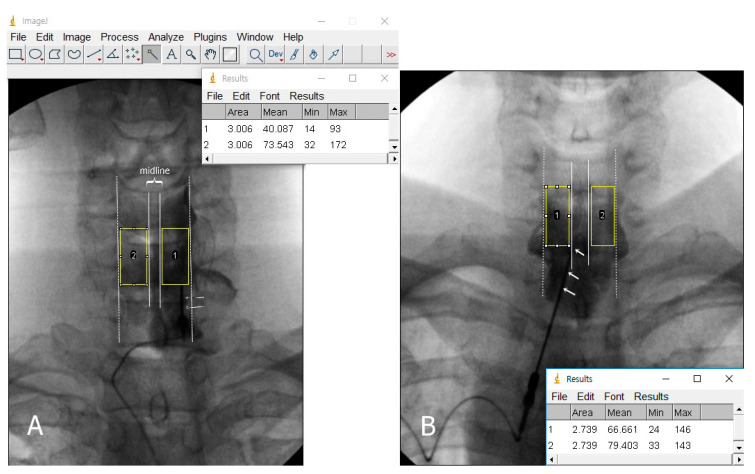
The grayscale brightness results of rectangular region of interest calculated by the ImageJ software. (**A**) Paramedian approach. The ratio was calculated by 73.54 (area 2)/40.08 (area 1), which is 1.83. This may indicate that a larger amount of injectate containing contrast spread in the ipsilateral side (area 1). (**B**) Midline approach. The ratio was calculated by 79.40 (area 2)/66.66 (area 1), which is 1.19. Dotted lines indicate pedicle lines. 1, ipsilateral side; 2, contralateral side; arrow, the final needle position.

**Figure 3 medicina-57-00008-f003:**
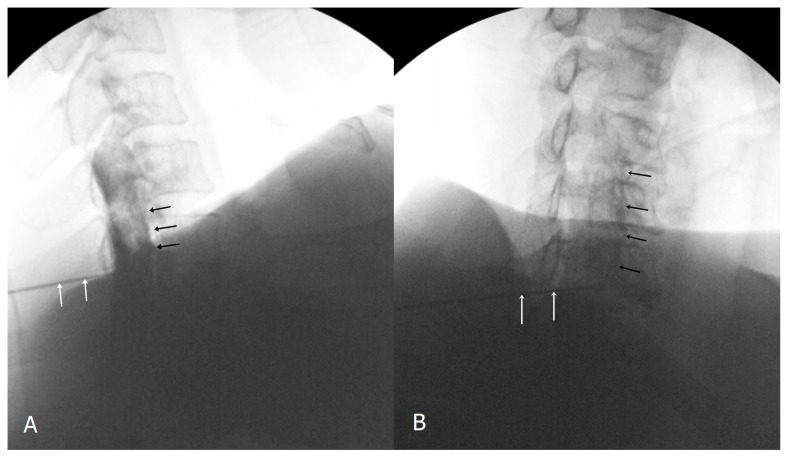
Confirmatory contrast medium spread of ventral epidural space in lateral and contralateral view (black arrow). White arrows indicate Tuohy needle. (**A**), lateral view; (**B**), contralateral view.

**Figure 4 medicina-57-00008-f004:**
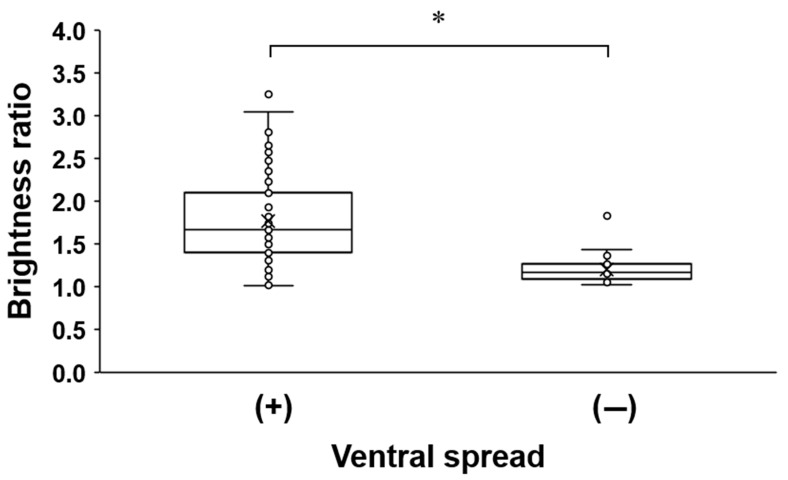
Association between ventral spread and grayscale brightness ratio. * *p* < 0.001.

**Figure 5 medicina-57-00008-f005:**
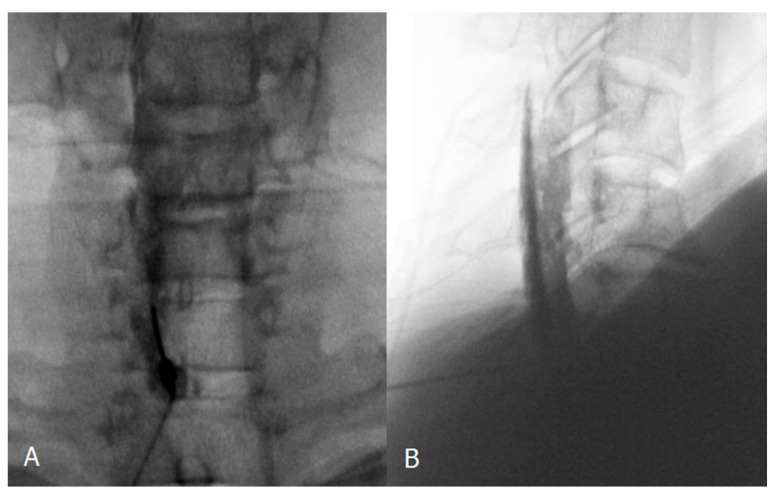
Confirmatory contrast medium spread of cervical interlaminar epidural injection. Anteroposterior (**A**) and lateral (**B**) cervical epidurogram showing left paramedian approach at C6-7 level. Contrast medium spreads predominantly longitudinally and bilaterally, but it did not reach to the ventral epidural space.

**Table 1 medicina-57-00008-t001:** Patients’ baseline and clinical characteristics.

.		Group					
Variable	Total (*n* = 84)	Group M (*n* = 42)	Group P (Pm + Pl)				*p* *
			Overall (*n* = 42)	Group PI (*n* = 16)	Group Pm (*n* = 26)	*p* ^†^	
Sex							
male	48 (57.1)	20 (47.6)	28 (66.7)	11 (68.8)	17 (65.4)	0.822 ^1^	0.078 ^1^
female	36 (42.9)	22 (52.4)	14 (33.3)	5 (31.3)	9 (34.6)		
Age (years)	57.02 ± 12.47	57.93 ± 12.97	56.12 ± 12.04	57.19 ± 11.46	55.46 ± 12.56	0.658 ^3^	
Impression							
HIVD	48 (57.1)	24 (57.1)	24 (57.1)	7 (43.8)	17 (65.4)	0.169 ^1^	10.000 ^1^
NF stenosis	29 (34.5)	15 (35.7)	14 (33.3)	7 (43.8)	7 (26.9)	0.261 ^1^	0.818 ^1^
Cervical sprain	4 (4.8)	1 (2.4)	3 (7.1)	1 (6.3)	2 (7.7)	1.000 ^2^	0.616 ^2^
PHN	3 (3.6)	2 (4.8)	1 (2.4)	1 (6.3)	0 (0.0)	0.381 ^2^	1.000 ^2^
Level							
C5-C6	7 (8.3)	1 (2.4)	6 (14.3)	2 (12.5)	4 (15.4)	0.909 ^2^	<0.001 ^2^
C6-C7	23 (27.4)	5 (11.9)	18 (42.9)	8 (50.0)	10 (38.5)		
C7-T1	54 (64.3)	36 (85.7)	18 (42.9)	6 (37.5)	12 (46.2)		

Values are either frequency with percentage in parentheses or mean ± standard deviation; ^1^
*p* values were derived by chi-squared test; ^2^
*p* values were derived from Fisher’s exact test; ^3^
*p* values were derived from independent *t*-test; * *p* values were derived from comparison between Group M (*n* = 42) and Group P (*n* = 42); ^†^
*p* values were derived from comparison between Group PI (*n* = 16) and Group Pm (*n* = 26); Shapiro–Wilk’s test was employed for test of normality assumption. Abbreviations: HIVD, herniated intervertebral disc; NF, neural foraminal; PHN; postherpetic neuralgia.

**Table 2 medicina-57-00008-t002:** Comparison of grayscale brightness ratio and incidence of ventral spread between Group M and Group P.

		Group		
Variable	Overall (*n* = 84)	Group M (*n* = 42)	Group P (Pm + Pl) (*n* = 42)	*p* Value
Brightness ratio	1.47 (1.17–1.94)	1.17 (1.10–1.41)	1.91 (1.50–2.35)	<0.001 ^1^
Ventral spread				
(+)	61 (72.6)	24 (57.1)	37 (88.1)	0.001 ^2^
(−)	23 (27.4)	18 (42.9)	5 (11.9)	

Values are either frequency with percentage in parentheses or median (IQR); ^1^
*p* values were derived from Mann–Whitney’s U test; ^2^
*p* values were derived by chi-squared test; Shapiro–Wilk’s test was employed for test of normality assumption.

**Table 3 medicina-57-00008-t003:** Comparison of grayscale brightness ratio and incidence of ventral spread between Group Pl and Group Pm.

		Group		
Variable	Overall (*n* = 42)	Group Pl (*n* = 16)	Group Pm (*n* = 26)	*p* Value
Brightness ratio	1.94 ± 0.53	1.92 ± 0.50	1.94 ± 0.55	0.912 ^1^
Ventral spread				
(+)	37 (88.1)	15 (93.8)	22 (84.6)	0.633 ^2^
(−)	5 (11.9)	1 (6.3)	4 (15.4)	

Values are either frequency with percentage in parentheses or mean ± standard deviation; ^1^
*p* values were derived from independent *t*-test; ^2^
*p* values were derived from Fisher’s exact test; Shapiro–Wilk’s test was employed for test of normality assumption.

**Table 4 medicina-57-00008-t004:** Association between ventral spread and grayscale brightness ratio.

		Ventral Spread		
Variable	Overall (*n* = 84)	(+) (*n* = 61)	(−) (*n* = 23)	*p* Value
Brightness ratio	1.47 (1.17–1.94)	1.66 (1.41–2.10)	1.16 (1.09–1.26)	<0.001 ^1^

Values are median (IQR); ^1^
*p* values were derived from Mann–Whitney’s U test; Shapiro–Wilk’s test was employed for test of normality assumption.

## Data Availability

The data presented in this study are available on request from the corresponding author.

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
