# Peer review of "Evaluation of Contrast Flow Patterns with Cervical Interlaminar Epidural Injection: Comparison of Midline and Paramedian Approaches"

_medicina, 2020, doi:10.3390/medicina57010008_

Round 1
Reviewer 1 Report
page 10 line 259: need some references in this paragraph
figure 2 the asterisk should be explained
Please explain how differentiation between group pm and group pi was made for the study
Author Response
Thank you for your kind comments. We got a lot of help from you in revising my manuscript. We revised my manuscript according to your opinions.
We appreciate your reviewing and look forward to receiving further response
page 10 line 259: need some references in this paragraph
Response: We added references in this paragraph. Page 9-10, line 253-254.
figure 2 the asterisk should be explained
Response: We added an explanation. Page 7, line 173.
Please explain how differentiation between group pm and group pi was made for the study
Response: We described this content. Page 2, line 73-76.

Reviewer 2 Report
The authors retrospectively showed that the cervical epidural injection of the contrast delivered by the paramedian approach could spread in the ipsilateral side more strongly than delivered by the median approach using ImageJ. However, there are some concerns.
Introduction
It is unclear about what is known, what is unknown. It is also unclear the present problem and/or purpose (and/or aim) of the present study. The authors should clarify these points.
I think that some topics can be mentioned briefly in the introduction section rather than in the discussion section, for example, the 3rd paragraph "CIEI is widely used to treat ...". I feel that the introduction is too short and the discussion is too long and redundant.
Methods
How did the practitioner decide or select the approach, midline, or paramedian? Were there some criteria? Did It depend on the patient's condition? Please clarify because it could be a bias.
How did the author define Pm and Pl? How median or lateral would be Pm or Pl? Please clarify although the differences between the two groups seemed to be unclear in the results.
Lines 64-66: The number, such as "(6)" is the number of the patients? or citations? If necessary, you had better show a figure, such as a flow diagram.
Results
I would like to know the clinical outcomes. If the contrast spread the ipsilateral side more strongly, the unilateral effect could be observed in the patients after the epidural injection. This is clinically important for the patients who suffered from chronic pain to relieve the pain without affecting the healthy side. If the practitioners recorded clinical findings such as the anesthetized area or analgesic effect of CEI of local anesthetics, this study would have more implications to show the clinical outcomes.
The authors included 10 patients who had two injections. The contrast would spread the same way when it was performed repeatedly in the same patient. I think that these patients should be excluded from the analysis because it could cause a bias.
The author showed the p-value in the tables adequately, but in the manuscript, just only "p<0.05". Please show the exact values such as p<0.001 or p=0.10.
Lines 116-117: the authors said each group was comparable, but I found a difference in the sex although I don't think that it would affect the results. How did you think about the p=0.04, although this might be a minor problem?
Figure: I would like to see the difference in the spread of contrast media between paramedian and midline approach. Please show the figure of the midline approach. This would help readers to understand the results.
Table 1: When using abbreviations, please indicate the non-abbreviated words as well. I could not understand the abbreviations (HIVD? NF? PHN may be postherpetic neuralgia?)
Tables: The authors used a non-parametric test for statistical analysis, but data are shown by mean (SD). In this case, I think that the median with the interquartile range or range is more adequate.
Discussion
In this section, the authors should simply discuss based on the findings from the results. This section was too long and redundant. I ask the authors to describe this section more simply.
As mentioned above, some topics should be shown in the introduction section rather than the discussion section, such as anatomical background and the related problem and clinical importance of CEI. I feel that there were several topics included in one paragraph.
I would like to know the clinical implications of this study. What can you show for us clinicians based on the results of this study? What is the benefit for the patients?
There were many author names written such as "XX et al. reported", but the content of the paper is important, not the author's name. Most readers would not be interested in the names.
Others
Funding: The template is described and it does not seem to be accurate.
Author Response
Response to Reviewer 2 Comments
Thank you for your kind comments. We got a lot of help from you in revising my manuscript. We considered your opinions as much as possible. We did our best to revise my manuscript. We appreciate your reviewing and look forward to receiving further response.
Introduction
It is unclear about what is known, what is unknown. It is also unclear the present problem and/or purpose (and/or aim) of the present study. The authors should clarify these points.
I think that some topics can be mentioned briefly in the introduction section rather than in the discussion section, for example, the 3rd paragraph "CIEI is widely used to treat ...". I feel that the introduction is too short and the discussion is too long and redundant.
Response: We revised the introduction section. According to your point, some topics in the discussion were mentioned briefly in the introduction. We tried to clarify the present problem and purpose of the present study.
Problem: To our knowledge, no consensus is available regarding the superiority of either approach for management of cervical spinal pain.
Purpose: The purpose of this study was to detail evidence showing the differences between the two approaches for contrast media and therapeutic injectate in reaching the suspected site of the origin of cervical pain during CIEI.
Page 1-2, line 30-57.
Methods
How did the practitioner decide or select the approach, midline, or paramedian? Were there some criteria? Did It depend on the patient's condition? Please clarify because it could be a bias.
Response: We revised the injection technique section. Page 2, line 82-83.
How did the author define Pm and Pl? How median or lateral would be Pm or Pl? Please clarify although the differences between the two groups seemed to be unclear in the results.
Response: We described this content. Page 2, line 73-76.
Lines 64-66: The number, such as "(6)" is the number of the patients? or citations? If necessary, you had better show a figure, such as a flow diagram.
Response: We added a flow diagram, Figure 1. Page 6.
Results
I would like to know the clinical outcomes. If the contrast spread the ipsilateral side more strongly, the unilateral effect could be observed in the patients after the epidural injection. This is clinically important for the patients who suffered from chronic pain to relieve the pain without affecting the healthy side. If the practitioners recorded clinical findings such as the anesthetized area or analgesic effect of CEI of local anesthetics, this study would have more implications to show the clinical outcomes.
Response: Thank you for your opinion. I agree with your point. We think the analysis of clinical findings can improve the quality of our report. But our study was retrospective design. We did not have uniform clinical findings. We’re afraid we can’t suggest quantitative analysis results of clinical findings in this study.
According to our literature review, there were many studies that conducted only contrast spread pattern analysis in the field of fluoroscopic guided interventions. (ref. 6,7,15 in our manuscript). Our study can provide detailed contrast spread data for the value of a paramedian approach for CIEI. We believe that further prospective research is needed based on our results. We also described this point as a limitation. Page 9, line 249-256.
The authors included 10 patients who had two injections. The contrast would spread the same way when it was performed repeatedly in the same patient. I think that these patients should be excluded from the analysis because it could cause a bias.
Response: I agree with your opinion. We corrected subjects and performed statistical analysis again. We included 84 cases. We revised our manuscript. Tables, Result
The author showed the p-value in the tables adequately, but in the manuscript, just only "p<0.05". Please show the exact values such as p<0.001 or p=0.10.
Response: We revised this point. Result part. Tables, Result
Lines 116-117: the authors said each group was comparable, but I found a difference in the sex although I don't think that it would affect the results. How did you think about the p=0.04, although this might be a minor problem?
Response: This flaw has been corrected by new data. Page 5, Table 1
Figure: I would like to see the difference in the spread of contrast media between paramedian and midline approach. Please show the figure of the midline approach. This would help readers to understand the results.
Response: We modified figure 2-B. Page 6
Table 1: When using abbreviations, please indicate the non-abbreviated words as well. I could not understand the abbreviations (HIVD? NF? PHN may be postherpetic neuralgia?)
Response: We added the abbreviations. Page 5, Table 1, line 142-143
Tables: The authors used a non-parametric test for statistical analysis, but data are shown by mean (SD). In this case, I think that the median with the interquartile range or range is more adequate.
Response: We added median values in the Table 2, 4 that were derived from Mann-Whitney’s U- test. Page 5, Table 2, 4.
Discussion
In this section, the authors should simply discuss based on the findings from the results. This section was too long and redundant. I ask the authors to describe this section more simply.
As mentioned above, some topics should be shown in the introduction section rather than the discussion section, such as anatomical background and the related problem and clinical importance of CEI. I feel that there were several topics included in one paragraph.
Response: We rearranged the discussion section according to your comment.
- Key results
- Interpretation
- contrast spread patten using ImageJ (unilateral tendency)
- the presence of ventral epidural spread
- Limitations
- Conclusions
I would like to know the clinical implications of this study. What can you show for us clinicians based on the results of this study? What is the benefit for the patients?
Response: We revised the conclusion section. Page 10, line 274-275.
We suggest that the paramedian approach can be more effective than the midline approach for the conservative management of unilateral cervical spinal pain. Our study provides the evidence for the practitioner to perform a more effective approach. It should be considered to improve patient’s outcomes.
There were many author names written such as "XX et al. reported", but the content of the paper is important, not the author's name. Most readers would not be interested in the names.
Response: We modified 3 paragraphs of descriptions such as "XX et al. reported"
Page 8, line 189-193.
Others
Funding: The template is described and it does not seem to be accurate.
Response: We described it correctly. Page 10, line 281.

Round 2
Reviewer 2 Report
First of all, the revised manuscript was very difficult to read because it reflected a history of changes, and the number of lines was different from what the author had written, so it was difficult to indicate the appropriate number of lines.
I think that the manuscript was improved. It is unfortunate that clinical outcome data cannot be used, but I understood that this was unavoidable due to the nature of the study.
I will add some comments below.
2.2 Injection technique
If the strategy depends on the practitioner's preference or the patient's conditions, these factors would affect the results and might cause some bias. It would be better to note it in Limitations, including the fact that the present study was a retrospective study.
In the Tables, there is a section where both the median (IQR) and the mean ± SD are written, but there is no reason to write both, and the mean seems unnecessary.
Figures: There are several Figures without titles. Are those unnecessary?
Figure 2: In the figure legends, it should be clearly shown that A is the paracentral approach and B is the median approach.
Funding: not "fundingor", but maybe "fundings"?
Author Response
Response to Reviewer 2-2 Comments
Thank you for your concerns. We found your comments helpful in structuring our manuscript for submission and in designing future studies.
2.2 Injection technique
If the strategy depends on the practitioner's preference or the patient's conditions, these factors would affect the results and might cause some bias. It would be better to note it in Limitations, including the fact that the present study was a retrospective study.
Response: We revised the limitation part. According to your point, the limitation was mentioned.
Pages 9, lines 250-251
In the Tables, there is a section where both the median (IQR) and the mean ± SD are written, but there is no reason to write both, and the mean seems unnecessary.
Response: We eliminated mean ± SD in the Table 2, 4.
Pages 5, Table 2, 4
Figures: There are several Figures without titles. Are those unnecessary?
Response: We added a title in the figure 5. We think those figure legends were enough to explain the figures.
Pages 8, lines 176
Figure 2: In the figure legends, it should be clearly shown that A is the paracentral approach and B is the median approach.
Response: We agree with your opinion. We modified the figure legend.
Pages 6, lines 164, 166
Funding: not "fundingor", but maybe "fundings"?
Response: We revised this point.
Pages 10, lines 283